# Comparable Overall Survival in Patients with Hepatocellular Carcinoma Diagnosed within and outside a Surveillance Programme: The Potential Impact of Liver Cirrhosis

**DOI:** 10.3390/cancers15030978

**Published:** 2023-02-03

**Authors:** Rosemary E. Faulkes, Zaira Rehman, Swetha Palanichamy, Nekisa Zakeri, Chris Coldham, Bobby V. M. Dasari, M. Thamara P. R. Perera, Neil Rajoriya, Shishir Shetty, Tahir Shah

**Affiliations:** 1Liver Unit, Queen Elizabeth Hospital, Birmingham B15 2GW, UK; 2Institute Immunology and Immunotherapy, University of Birmingham, Birmingham B15 2TT, UK

**Keywords:** hepatocellular carcinoma, surveillance, cirrhosis, tumour microenvironment

## Abstract

**Simple Summary:**

Screening for liver cancer (hepatocellular carcinoma, HCC) is recommended for people with liver cirrhosis and some people with chronic liver disease. This study compares outcomes between patients diagnosed with HCC through surveillance and patients not included in surveillance. Patients diagnosed with HCC through surveillance had smaller tumours and an earlier cancer stage, but also had a higher incidence of cirrhosis. Although treatment rates were similar between the two groups, there was no difference in survival. This highlights the impact that liver cirrhosis has on tumour behaviour in HCC.

**Abstract:**

Hepatocellular carcinoma (HCC) is the third leading cause of cancer death, and its incidence is rising. Mortality from HCC is predicted to increase by 140% by 2035. Surveillance of high-risk patients with cirrhosis or chronic liver disease may be one means of reducing HCC mortality, but the level of supporting evidence for international guidelines is low/moderate. This study explores the real-world experience of HCC surveillance at a tertiary referral centre. Electronic patient records for all new HCCs diagnosed between August 2012 and December 2021 were retrospectively reviewed. Patient and tumour characteristics were evaluated, including the co-existence of chronic liver disease, cancer treatment and survival, and categorised according to HCC diagnosis within or outside a surveillance programme. Patients with HCC who presented through surveillance had smaller tumours diagnosed at an earlier stage, but this did not translate into improved overall survival. All patients in surveillance had chronic liver disease, including 91% (*n* = 101) with cirrhosis, compared to 45% (*n* = 29) in the non-surveillance cohort. We propose that the immune dysfunction associated with cirrhosis predisposes patients to a more aggressive tumour biology than the largely non-cirrhotic population in the non-surveillance group.

## 1. Introduction

Hepatocellular carcinoma (HCC) is the third leading cause of cancer-related death worldwide, and the incidence is rising annually in Europe, the USA and Australia [1]. Five-year survival in the UK is extremely poor, at less than 15% [2]. Whilst overall cancer survival has improved over recent decades, mortality from HCC shows the opposite trend and has risen by 45% over the course of only 12 years [3]. By 2035, deaths from HCC are predicted to increase by over 132% for women and 140% for men [4]. This is due to the rising prevalence of chronic liver disease—in particular, alcohol-related liver disease and non-alcoholic steatohepatitis (NASH). Additionally, curative treatment options for HCC are limited by the compromised state of the liver parenchyma and the immunosuppressive microenvironment contributing to tumour immune escape [5]. 

In the majority of cases, HCC develops in people with underlying cirrhosis or pre-cirrhotic patients at high risk, such as certain patients with chronic hepatitis B infection. Early identification and treatment of HCCs may be one means of improving cancer survival, and these cohorts meet the criteria for a survival benefit from cancer screening [6]. Current international guidelines from the European Association for the Study of the Liver (EASL) and the American Association for the Study of Liver Disease (AASLD) recommend six-monthly surveillance ultrasound imaging, with or without serum alpha foetoprotein (AFP) measurement [6,7]. Although the level of recommendation is strong, the level of evidence is low/moderate. 

There are a number of areas of debate regarding current surveillance practices and the feasibility of implementing guidelines. For example, EASL recommends a liver biopsy for indeterminate nodules that cannot be diagnosed radiologically, although the size and location of liver lesions can often mean that biopsy is not technically feasible [8]. Another challenge relates to the use of ultrasound, which is less sensitive in those with a body mass index (BMI) > 30 kg/m^2^ [9]. As obesity reaches endemic proportions, there is an ongoing debate about the most appropriate modality for HCC surveillance that is both clinically and cost-effective. Alternative screening protocols using non-contrast magnetic resonance imaging (MRI) may be more sensitive in this population, although further economic evaluation is required [10]. 

Between 11 and 19% of patients who develop HCC do not have cirrhosis and are therefore not included in surveillance programmes [11,12]. Another difficulty facing HCC surveillance relates to NASH, which has a 5% incidence in the United States. A small proportion of non-cirrhotic patients with NASH will develop HCC, most likely those with advanced fibrosis [13]. More work is needed to develop a means of identifying patients at risk of developing HCC within this cohort. 

Despite these areas of controversy, a recent meta-analysis has shown that surveillance for HCC does increase overall survival after adjustment for lead-time bias [14]. However, there are minimal data on the real-world experience of differences in survival between surveillance and non-surveillance groups in HCC. 

The Birmingham Liver Unit operates a large tertiary referral HCC service in conjunction with HCC surveillance for at-risk populations with liver disease. A retrospective analysis of patients presenting within and outside an HCC surveillance programme was performed, with a hypothesis that patients under HCC surveillance would be diagnosed with earlier-stage tumours and therefore have a better prognosis. The aim of this study was to confirm the hypothesis by comparing patient and tumour demographics and survival outcomes for patients diagnosed at our centre within and outside surveillance, and to accurately assess overall survival and determine potential prognostic factors between the two groups. The authors also sought to clarify why patients that met surveillance criteria were not enrolled in the surveillance programme, in order to identify areas for improvement within the HCC service and increase screening rates. 

## 2. Materials and Methods

### 2.1. Study Design

A retrospective review of patient records from an electronic hospital database at the Liver Unit, Queen Elizabeth Hospital, Birmingham, UK was performed. The study gained local audit approval (CARMS no. 17979). Inclusion criteria were patients over 18 years old diagnosed with a new HCC at this institution between August 2012 and December 2021 (Figure 1). HCC diagnosis was determined at the tertiary centre multi-disciplinary team meeting, including an experienced specialist hepato-biliary radiologist, using multiphase imaging (computerised tomography, CT, or magnetic resonance imaging, MRI), or histology as per EASL and AASLD guidelines [6,7]. Diagnosis of chronic liver disease or cirrhosis was recorded according to previous diagnosis in patient records. For patients without an established diagnosis of liver disease prior to HCC presentation, records were searched for radiological and biochemical evidence of cirrhosis (coarse, irregular liver on imaging, splenomegaly, varices, ascites or thrombocytopenia) or data consistent with chronic liver disease (history of metabolic syndrome or positive hepatitis B or C serology). External hospital referrals were not included, due to inability to accurately assign these patients to the two study groups or guarantee that all HCC diagnoses were referred. Patients were categorised according to whether HCC was diagnosed within a surveillance programme, or presenting outside surveillance with symptoms or an incidental finding on imaging. 

Data parameters collected included patient demographics, tumour characteristics (size and number) and blood results, including liver function and serum AFP. The World Health Organisation (WHO) performance status, a global assessment of a patient’s physical fitness and frailty, used to inform decisions regarding cancer treatment, was also recorded. Data were used to calculate the following liver prognosis scores: United Kingdom Model for End Stage Liver Disease (UKELD), Model for End Stage Liver Disease (MELD) and Childs Pugh score. UKELD and MELD assess the severity of liver disease using a composite of serum bilirubin, International Normalised Ratio (INR), sodium and creatinine. MELD also takes into account whether dialysis is used. Childs Pugh is used to prognosticate in cirrhosis, comprising bilirubin, clotting parameters and albumin, along with the presence or absence of ascites and hepatic encephalopathy. The HCC prognosis scores Duvoux and Barcelona Clinic Liver Cancer (BCLC) stage were also calculated [15,16]. BCLC stages HCC according to tumour number, size and Childs Pugh score. The Duvoux score is validated for use in evaluating the response to HCC treatment, and includes tumour size, number and AFP. Treatment given and overall patient survival were also recorded. 

### 2.2. Data Analysis

Data analysis was performed using SPSS Version 24 (2016) and GraphPad Prism 9.5.0 (525). Patient characteristics were summarised as proportions (%) for categorical variables and mean ± SD or median (IQR) for continuous variables. Comparison between groups was performed using chi-square. Overall survival analysis was carried out using Kaplan–Meier curves and difference between survival curves was evaluated using the log rank test. *p* values < 0.05 were deemed statistically significant.

## 3. Results

### 3.1. Patient and Tumour Characteristics

A total of 175 patients (male = 134, mean age 65) were diagnosed with HCC within the study period, of whom 63% (*n* = 111) were diagnosed through HCC surveillance and 37% (*n* = 64) were not (Table 1). 

There were no statistically significant differences in age or sex between the two groups. The surveillance cohort had a slightly lower mean body mass index (BMI) compared to non-surveillance (28 vs. 30.8, *p* ≤ 0.05) and a higher functional ability as categorised by WHO performance status (PS). Significantly more patients within surveillance were PS 0 than in the non-surveillance group (85%, *n* = 94 versus 58%, *n* = 37, *p* ≤ 0.05). In contrast, the incidence of frailty as measured by a need for nursing care (PS 3) was significantly higher in the non-surveillance cohort (9%, *n* = 6 of non-surveillance versus 2%, *n* = 2 for surveillance *p* ≤ 0.05) (Figure 2).

In the surveillance group, 91% (*n* = 101) of patients had cirrhosis, and the remaining 9% were under HCC surveillance due to individual cancer risk assessment based on the presence of chronic hepatitis B, Alagille Syndrome, glycogen storage disorder, autoimmune hepatitis or NASH. Within the non-surveillance group, 44 (69%) had chronic liver disease, of whom 29 patients (45%) also had cirrhosis, and the remaining 20 patients (31%) had no evidence of liver disease. Aetiology of liver disease is shown in Figure 3. Non-alcoholic steatohepatitis (NASH), alcohol and viral hepatitis were the leading causes of liver disease for both groups, although the distribution of these diseases varied. Incidence of alcohol-related liver disease (27%) and hepatitis C (27%) was significantly higher in the surveillance group, compared to 13% and 6% for non-surveillance, respectively (*p* < 0.05).

There was no difference in median AFP between the groups, but patients in the non-surveillance group had a significantly larger mean tumour size at diagnosis compared to the surveillance group (5.3 cm vs. 2.9 cm, *p* ≤ 0.05). This difference was reflected by the BCLC staging. Most patients in the surveillance group were BCLC stage A at diagnosis (84%, *n* = 93), compared to 64% (*n* = 41) in the non-surveillance group (*p* ≤ 0.05). Mean Duvoux score was also significantly different between the two groups (0 for surveillance, 1 for non-surveillance, *p* ≤ 0.05).

### 3.2. HCC Treatment and Survival

Patients were treated with current best available modalities as determined by multidisciplinary review. These included surgical resection, liver transplantation, radiofrequency ablation (RFA), transarterial chemoembolisation (TACE), stereotactic ablative radiotherapy (SABR) and systemic treatment with tyrosine kinase inhibitors (Sorafenib, Lenvatinib) or combined therapy with Atezolizumab and Bevacizumab (Figure 4). Some patients received more than one form of treatment. Eight patients (13%) were managed with best supportive care (BSC) in the non-surveillance group, comparable with 12 patients (11%) in the surveillance group. A higher proportion of patients in the surveillance group received ablation treatment (22% vs. 8% non-surveillance, *p* ≤ 0.05), in line with the smaller average tumour size within this group. In respect to surgical treatment, those diagnosed outside surveillance were more likely to be non-cirrhotic, and, as a result, the rate of liver resection was higher compared to surveillance (28% vs. 5% respectively, *p* ≤ 0.05), whereas the incidence of liver transplantation was greater in the surveillance group (11% vs. 3% in non-surveillance, *p* = 0.06).

Despite an earlier diagnosis of HCC through surveillance, as demonstrated by the smaller tumour size and earlier BCLC stage, there was no statistically significant difference in overall survival between the two groups. Median survival was 3.67 years in the surveillance group and 3.58 years in the non-surveillance group (*p* = 0.41, Figure 5).

### 3.3. Comparison of Cirrhotic and Non-Cirrhotic Patients in the Non-Surveillance Cohort

Within the non-surveillance cohort (*n* = 64), 35 patients (69%) had underlying chronic liver disease, of whom 33 were undiagnosed and presented with de novo cirrhosis and HCC. Two patients had a prior diagnosis of cirrhosis and could have been under surveillance; one patient had been lost to follow-up, and the other was diagnosed with cirrhosis in primary care but had not been referred to secondary care.

We performed a further analysis of tumour demographics between the cirrhotic and non-cirrhotic patients in the non-surveillance group (Table 2). Age, sex and BMI were similar in those with and without cirrhosis. In both groups, NASH was the commonest aetiology. Non-cirrhotic patients had a larger tumour size at diagnosis compared to patients with cirrhosis, mean 7 cm and 4.5 cm, respectively (*p* ≤ 0.05).

Overall survival within the non-surveillance group according to the presence or absence of cirrhosis was reviewed (Figure 6). Median survival was slightly higher in patients with cirrhosis, but this did not reach statistical significance (3.58 years vs. 2.83 years, *p* = 0.39).

## 4. Discussion

This study presents real-world data from a single centre over a ten-year period, demonstrating that patients under HCC surveillance were more likely to be diagnosed with small tumours at an earlier BCLC stage. They also had a more favourable WHO performance status compared to those diagnosed outside surveillance, and were more likely to undergo curative treatments of ablation or liver transplantation. However, despite these apparent advantages, we report no difference in overall survival between surveillance and non-surveillance populations. These findings are likely to reflect the high incidence of cirrhosis in the surveillance group, the implications that this has for treatment choice and the effect of cirrhosis on tumour biology.

Several differences in the genomic and immunological landscape have been described for HCC diagnosed in cirrhotic versus non-cirrhotic livers. While alterations of the p53 pathway have been implicated in the development of HCC in cirrhosis, alterations of cell cycle regulators p21 and p27 were found to play a greater role in the pathogenesis of non-cirrhotic HCC [17]. The immune microenvironment of cirrhosis might also predispose patients to a more aggressive tumour phenotype. Cirrhosis-associated immune dysfunction, driven by increased gut barrier permeability and bacterial translocation, leads to the excessive activation and subsequent exhaustion of the innate and adaptive immune system [18]. Upon the development of HCC, the further enrichment of negative regulatory immune cells, upregulation of inhibitory checkpoints on effector lymphocytes and elevated levels of inhibitory soluble molecules all contribute to a highly immunosuppressive tumour microenvironment, reducing the efficacy of endogenous anti-tumour immune responses [5,19]. The presence of cirrhosis can also render HCC more difficult to treat due to potential complications of hepatic decompensation, as well as increased risks of de novo HCC formation following treatment [20].

Our study is consistent with published literature comparing outcomes for HCC with and without cirrhosis. Between 11 and 19% of HCCs develop in the absence of cirrhosis, and NASH is the commonest cause of chronic liver disease in these cases [11,12,21]. Tumours in non-cirrhotic livers are usually larger, beyond Milan criteria at diagnosis and more likely to be treated with liver resection compared to cirrhotic HCC’s. Despite the adverse tumour characteristics in relation to size, overall survival has consistently been longer than survival in patients with cirrhosis and HCC. Gawrieh et al. report median survival for HCC of 1.3 years in cirrhosis and 1.8 years in non-cirrhosis in the US [12]. Van Meer et al. found that the absence of cirrhosis in a Dutch cohort was a significant predictor of improved survival, providing further evidence of the adverse effects of a cirrhotic environment in HCC outcomes [11]. Five-year survival for both groups in our study was better than the national average, which may be in part due to treatment at a high-volume, academic centre, which has previously been shown to improve HCC-related survival compared to low-volume centres [22].

These results differ from a recent meta-analysis that showed improved survival with HCC surveillance after adjustment for lead-time bias, although there was high heterogeneity between the 12 studies included [14]. Within the UK, a study from Scotland also found a survival advantage from surveillance (28.7 months versus six months) [23]. However, in these studies, the majority of cases were diagnosed outside surveillance. In Scotland, 70% presented outside surveillance, and in North America, non-surveillance diagnosis has been reported to be as high as 86% of all new HCCs [14,23]. The minority of HCCs that were diagnosed within surveillance therefore represent a select population group that may have better access to healthcare or different health beliefs manifesting as different health behaviours, and therefore confound the apparent survival advantage derived from surveillance [24,25]. In our study, the majority of HCCs were diagnosed through surveillance (64%), demonstrating the efficacy of the screening programme at our institution. Due to the more equitable sample distribution between the surveillance and non-surveillance groups, these results may also provide a more accurate evaluation of the effect of surveillance in HCC. This is also the only study to date to report survival in HCC surveillance in an English population.

Surveillance can appear to provide a survival advantage due to lead-time bias. Whilst we have not evaluated for this within our study, it is possible that survival in the surveillance cohort is thus affected. Cuchetti et al. identified a 7.2 month survival lead time for patients diagnosed with HCC through six-monthly surveillance [26]. However, when this was taken into account, there still remained a survival advantage with HCC surveillance, although incidental diagnoses were excluded. While our study evaluates a smaller number of patients, it more accurately reflects real-world experience by including incidental HCC diagnoses in the non-surveillance cohort.

These data are limited by being from a retrospective, single-centre study. Subgroup analysis in the non-surveillance group comparing the effect of cirrhosis with non-cirrhosis consists of a small patient sample, which may limit the power of the data. Further limitations relate to the use of the WHO performance status. Frailty in patients with cirrhosis is common but often under-diagnosed, and therefore the functional status of cirrhotic patients may not have been accurately categorised by the WHO performance status [27]. However, contemporaneous assessment of patients within our dataset included the WHO performance status only, which was therefore used in our analysis.

The COVID-19 pandemic has also adversely affected the diagnosis and treatment of HCC [8]. Our study period overlapped with the pandemic, but subgroup analysis comparing HCC outcomes before and during the pandemic was not performed as this was not the focus of our study. Furthermore, a medical team was assembled in our unit to provide routine outpatient care remotely, and thus HCC surveillance was maintained during the pandemic. A further limitation in evaluating the effect of surveillance is the absence of data pertaining to potential harms sustained—for example, the psychological effect of being enrolled in a surveillance programme—which was beyond the scope of this study.

## 5. Conclusions

HCC is often asymptomatic and, therefore, without surveillance, it is often diagnosed at a more advanced stage, as demonstrated in this study. However, despite smaller tumours and an earlier cancer stage at the time of diagnosis in the surveillance group, this cohort did not obtain a survival advantage. As these patients all had underlying liver disease, the presence of cirrhosis may have predisposed them to a more aggressive tumour biology, significantly impacting overall mortality despite achieving an earlier diagnosis through HCC surveillance.

## Figures and Tables

**Figure 1 cancers-15-00978-f001:**
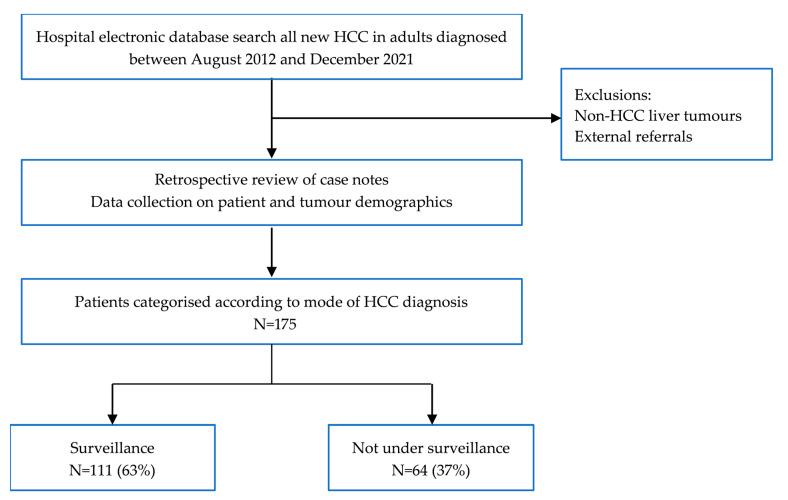
Patient selection for study.

**Figure 2 cancers-15-00978-f002:**
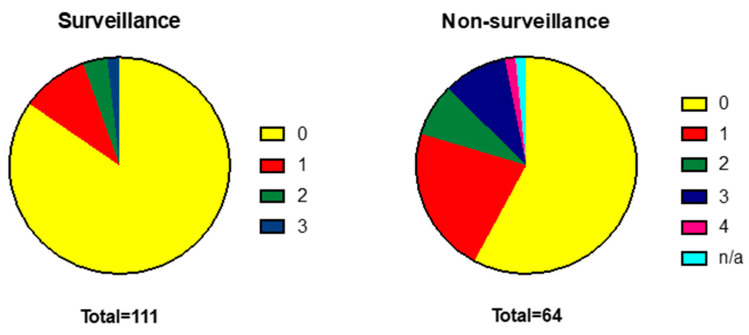
WHO performance status is more advanced in patients diagnosed outside a surveillance programme. WHO Performance Status: 0 = normal activity, 1—symptomatic and ambulatory, self-caring, 2—ambulatory >50% of time, requires occasional assistance, 3—ambulatory <50% of time, nursing care needed, 4—bedridden.

**Figure 3 cancers-15-00978-f003:**
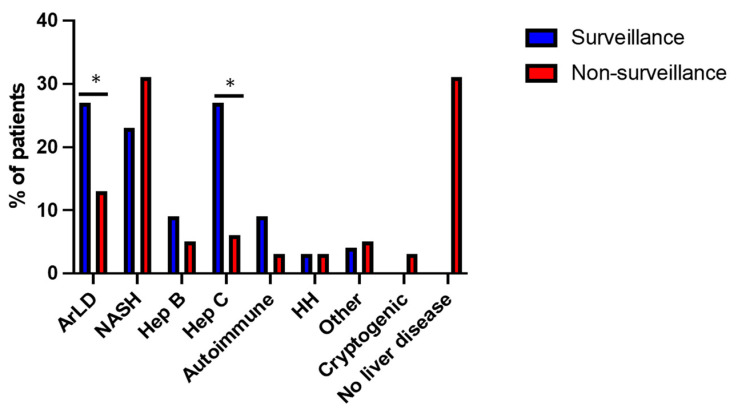
Aetiology of liver disease in surveillance and non-surveillance groups. Alcohol-related liver disease (ArLD), non-alcoholic steatohepatitis (NASH), hepatitis B (Hep B), hepatitis C (Hep C), hereditary haemochromatosis (HH), other—Alagille syndrome, glycogen storage disorder, congenital heart disease. * *p* ≤ 0.05 (chi-square test).

**Figure 4 cancers-15-00978-f004:**
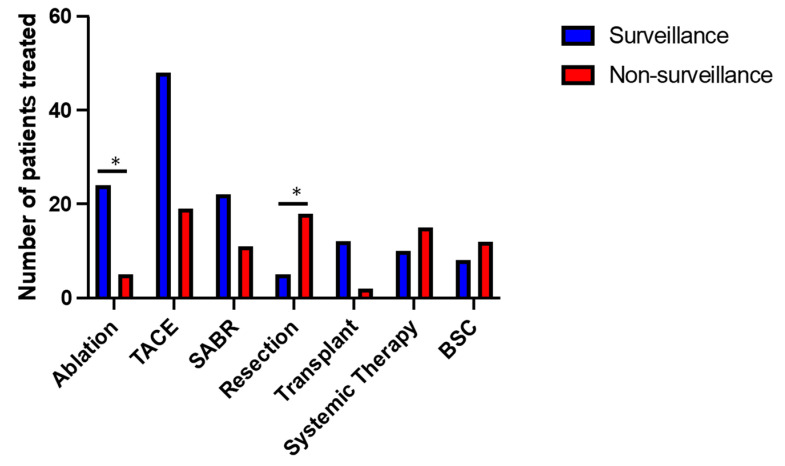
Overview of different treatments received in surveillance and non-surveillance groups. TACE (transarterial chemoembolisation), SABR (stereotactic ablative radiotherapy), BSC (best supportive care). * *p* < 0.05 (chi-square test).

**Figure 5 cancers-15-00978-f005:**
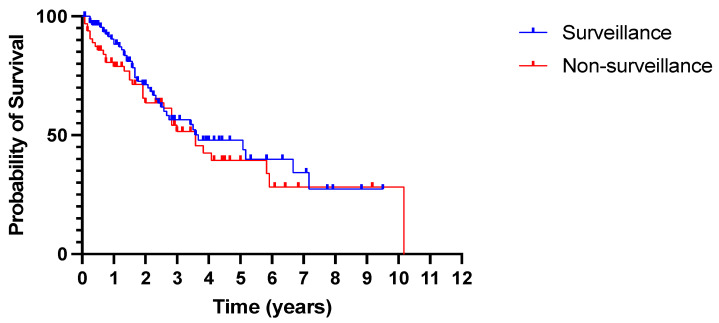
No difference in overall survival between patients diagnosed with HCC in and outside surveillance. All-cause mortality. *p* = 0.41 (log rank Mantel–Cox test).

**Figure 6 cancers-15-00978-f006:**
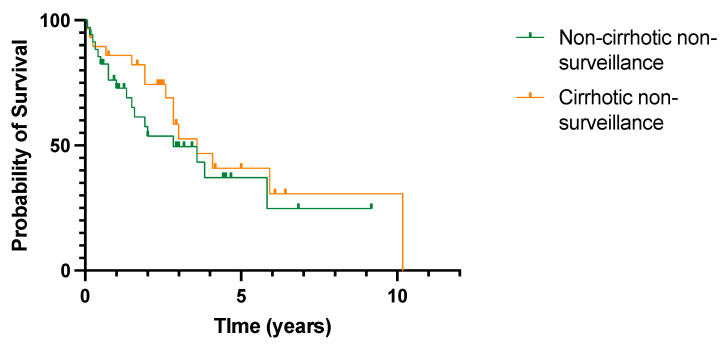
No difference in overall survival in patients with or without cirrhosis diagnosed with HCC outside cancer surveillance. All-cause mortality. *p* = 0.39 (log rank (Mantel–Cox) test).

**Table 1 cancers-15-00978-t001:** Patient and tumour demographics in surveillance and non-surveillance groups.

Demographics	Surveillance (*n* = 111)	Non-Surveillance (*n* = 64)	*p* Value
Age, years (range)	66 (59.6–73.1)	67 (24–87)	0.6
Male (%)	81 (73%)	53 (83%)	0.13
BMI median (IQR)	28 (24–34)	30.8 (19.3–46.9)	<0.05
WHO performance status 0 1 2 3 4 Unavailable	94 (85%) 11 (10%) 4 (3%) 2 (2%) 0 0	37 (58%) 14 (22%) 5 (8%) 6 (9%) 1 (1.5%) 1(1.5%)	<0.05 <0.05 0.14 <0.05 - -
Aetiology NASH Alcohol Hepatitis C Hepatitis B Autoimmune Haemochromatosis Cryptogenic Other No liver disease	25 (22.6%) 30 (27%) 30 (27%) 10 (9%) 10 (9%) 2 (1.8%) 0 4 (3.6%) 0	20 (31%) 8 (13%) 4 (6%) 3 (5%) 2 (3%) 2 (3%) 2 (3%) 3 (5%) 20 (31%)	0.25 <0.05 <0.05 0.34 0.13 0.83 - 0.76 -
Cirrhosis present	101 (91%)	29 (45%)	<0.05
Childs Pugh Score A B C	*n* = 101 91 (90%) 10 (10%) 0	*n* = 29 22 (76%) 7 (24%) 0	<0.05 <0.05 -
AFP median (IQR)	6 (3–36)	5 (3–25)	0.99
Douvoux score	0 (0–1)	1 (0–4)	<0.05
Size of largest tumour at diagnosis (cm)	2.9 (95%CI 2.5–3.3)	5.3 (95%CI 4.4–6.1)	<0.05
BCLC stage A B C D Unavailable	93 (83.8%) 14 (12.6%) 1 (0.9%) 1 (0.9%) 2 (1.8%)	41 (64%) 16 (25%) 5 (8%) 1 (1.5%) 1 (1.5%)	<0.05 0.07 0.23 0.77 0.58
HCC treatment * Ablation Liver transplant SABR Surgical resection Systemic therapy TACE Best supportive care	24 (22%) 12 (11%) 22 (20%) 5 (5%) 10 (9%) 48 (43%) 10 (9%)	5 (8%) 2 (3%) 11 (17%) 18 (28%) 15 (23%) 19 (30%) 12 (19%)	<0.05 0.06 0.62 <0.05 <0.05 0.1 0.06

* Some patients received more than one form of treatment. BMI (body mass index); NASH (non-alcoholic steatohepatitis); Other—Alagille syndrome, glycogen storage disorder or cirrhosis secondary to congenital heart disease; AFP (alpha foetoprotein); BCLC (Barcelona Clinic Liver Cancer stage), WHO (World Health Organisation) performance status: 0 = normal activity, 1—symptomatic and ambulatory, self-caring, 2—ambulatory > 50% of time, requires occasional assistance, 3—ambulatory < 50% of time, nursing care needed, 4—bedridden; SABR (stereotactic ablative radiotherapy); TACE (transarterial chemoembolisation).

**Table 2 cancers-15-00978-t002:** Patient and tumour demographics with and without cirrhosis in the non-surveillance cohort.

Demographics	Non-Surveillance No Cirrhosis (*N* =35)	Non-Surveillance Cirrhosis (*N* = 29)	*p* Value
Age (years) median (IQR)	72 (60–76)	68 (62–75)	0.22
Male (%)	27 (77%)	26 (90%)	0.17
BMI median (IQR)	30 (25–36)	31 (27–34)	0.88
Aetiology NASH Alcohol Hepatitis C Hepatitis B Hemochromatosis Autoimmune Other Cryptogenic No liver disease	9 (25.7%) 0 1 (2.9%) 0 1 (2.9%) 1 (2.9%) 3 (8.6%) 0 20 (57%)	11 (37.9%) 8 (27.6%) 3 (10.4%) 3 (10.4%) 1 (3.4%) 1 (3.4%) 0 2 (6.9%) 0	0.06 <0.05 0.25 0.06 1 1 - - -
AFP median (IQR)	5 (2–18)	7 (3–42)	0.41
Douvoux score	2.5 (0–4)	1 (0–3)	0.2
Size of largest tumour at diagnosis (cm)	7 (95%CI 5.6–8.4)	4.5 (95%CI 3.6–5.4)	<0.05

## Data Availability

The data presented in this study are available on request to the corresponding author.

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
