# Peer review of "Comparable Overall Survival in Patients with Hepatocellular Carcinoma Diagnosed within and outside a Surveillance Programme: The Potential Impact of Liver Cirrhosis"

_cancers, 2023, doi:10.3390/cancers15030978_

Round 1
Reviewer 1 Report
Manuscript (2151211): Comparable overall survival in patients with hepatocellular carcinoma diagnosed within and outside a surveillance programme: the potential impact of the tumor microenvironment
This manuscript compares out-comes between patients diagnosed with hepatocellular carcinoma (HCC) through surveillance versus patients not included in surveillance. Authors compare the aetiology of the liver disease, tumors size, cancer stages, cirrhosis incidence, survival and treatments between the two groups.
This statistical study is relevant for the diagnostic and prognosis of HCC. This study analyzed a relative good number of patients for each group. An extensive discussion was included. However, this work will need minor modifications.
Concerns and suggestions are listed below.
1. Title must be modified, since it does not reflect what the authors found. (Change “the potential impact of the tumor microenvironment” with a sentence that summarize you findings).
2. The result section is poorly written, you will need to improve it and be more explicit by describing the aim of each approach and the importance of your findings.
3. Materials and Methods section must briefly describe each liver prognosis scores.
4. In Materials and Methods, authors must include the meaning of all abbreviations (CT, MRI, WHO, BMI, etc.).
5. Figures numbers need to be changed (There are two figures 1 and 2, instead of figure 3 and 4).
6. In figure 1, there is a discrepancy between value in the graph and the table for HH in Surveillance group (in graph is indicated 2% and table indicates 3%)
7. In figures legends, authors must add figure legend title and all information necessary for understanding the figure. Including type of comparison, P values (page 7) and meaning of all abbreviations (Dx).
8. Add vertical SD bars on graphs.
9. Correction of “Error! Reference source not found”
10. The text in page 5, must make a reference about figure 2 presented in page 6.
Reviewer 2 Report
In their manuscript the authors present the outcomes from a retrospective single-center study aiming to compare patient and tumour demographics and survival outcomes for patients newly diagnosed with HCC at their centre within and outside surveillance, and to accurately assess overall survival and determine potential prognostic factors between the two groups.
This is an excellent study with a significant clinical message. The study is well-structured and the manuscript well-written. The study limitations are adequately acknowledged by the authors. The relevant literature is cited. The methodology is clearly presented as are the results.
Major points:
- Do the authors feel that recurrence rates should be reported for each group? It would be interesting to see if a higher recurrence rate within the cirrhotic population (which is expected) actually makes the non-surveillance group having a more aggressive biologically disease despite non-recurring.
Reviewer 3 Report
Dear Authors
I would like to thank you for the opportunity of reviewing this interesting paper that is focused on a very remarkable and challenging topic that is a lively argument also in the daily clinical practice.
Screening for liver cancer, especially for hepatocellular carcinoma (HCC), is recommended for people with liver cirrhosis and some people with chronic liver disease. This study compares outcomes between patients diagnosed with HCC through surveillance with patients not included in surveillance. Patients diagnosed with HCC through surveillance had smaller tumors and earlier cancer stage, but also had higher incidence of cirrhosis. Although treatment rates were similar between the two groups, there was no difference in survival. This highlights the impact liver cirrhosis has on tumor behavior in HCC.
Therefore, papers that explore in depth this theme, that always represented a great challenge for all interventional radiologist, but secondarily also for hepatologists, especially in the era of tailored medicine, could surely be of interest for this important journal. Moreover, this paper demonstrates the aim of finding objective and practical conclusions from the many studies that have been conducted in recent years.
SPECIFIC COMMENTS
This manuscript is pleasurable and flowing; no major issues are appreciable.
Although language used is appropriate, I (I am not a native English speaker) recommend to Authors to obtain a certified native speaker with proficiencies in the scientific-medical field to complete properly this paper (if not jet done). Moreover, I recommend making a further revision of the manuscript to fix some small typing/language errors.
TITLE
The title is clear and direct. Personally, I believe it could be improved and be more focused on the results, for example:“Comparable overall survival in patients with hepatocellular carcinoma diagnosed within and outside a surveillance programme: the potential impact of hepatic cirrhosis”. If the editor agrees, the authors could consider the change of the title.
ABSTRACT
The abstract is well structured in all its section and, therefore, it properly reflects the main text highlighting the most important aspects of this paper. Consequently, adjustments are not needed.
INTRODUCTION
First of all, despite the sentence “Five-year survival in the UK is extremely poor at less than 15%” seems appropriate, in order to more correctly quantify the burden of HCC it would be useful to provide also some worldwide data [doi: 10.3389/fonc.2020.00171].
Secondly, although the introduction fits the context of the study, it is concise. Sometime, many concepts clearly explicated in an exhaustive introduction could help readers to become passionate about reading the paper and using it as a reference. For example: “Early identification and treatment of HCCs may be one means of improving cancer survival.”. I believe this topic should be expanded. In fact, to facilitate HCC diagnosis, many new protocols are emerging, as testified by several studies regarding the potential use of non-contrast MRI as alternative surveillance tool for HCC in selected patients with the aim to significantly save costs and make it a more cost-effective strategy. Similarly, also several new imaging methodologies are being studied, including the administration of new PET-radiotracers and radiomics [doi: 10.1007/s11547-022-01449-w].
In addition, in the second paragraph “Current international guidelines from the European Association for the Study of the Liver and the American Association for the Study of Liver Disease recommend six monthly surveillance ultrasound imaging, with or without serum alfa foetoprotein (AFP) measurement.[5] [6] Although the level of recommendation is strong, the level of evidence is low/moderate.” it is important to underline that both European and American Guidelines have also several limitations. For example, despite both guidelines recommend histological confirmation for all focal liver lesion in cirrhotic patients with an atypical radiologic vascular pattern at CT and/or MRI, biopsy has several limitations (including patient constitution, tumour location, and differing operator skills) and a recent study demonstrated that its actual feasibility in these scenarios is deemed possible in slightly more than half of the cases, even by experienced radiologists. Therefore, a change in the diagnostic algorithm is warranted. [doi: 10.3390/jcm11154399]
MATERIALS & METHODS
In the section, 2.1. Study design, regarding the sentence “HCC diagnosis was determined by radiology (CT or MRI) or histology, and corroborated by a multi-disciplinary team.”. What imaging features did you consider for diagnosis? What CT and MRI protocol did you use? Please cite both European and American guidelines if imaging studies and HCC diagnosis was performed accordingly [doi: 10.1002/hep.29086; doi: 10.1016/j.jhep.2018.03.019]. Moreover, the experience of the radiology should be mentioned (for example, >10 years).
Please provide a flow chart of the selection of patients in order to facilitate readers’ comprehension.
RESULTS
Results remain clear and well-structured, and no major adjustments are needed.
In the second paragraph “Stage of liver disease as determined by presence of cirrhosis and Childs Pugh score was significantly higher in the surveillance group, where the majority of patients had Childs Pugh A cirrhosis (90%, n=91) compared to the non-surveillance group.”, the sentence is not very clear and it might confuse readers as it has been written, thus I would stick to saying that the group under surveillance had a higher percentage of patients with cirrhosis compared to the other group, with the majority of them at the Child Pugh A stage.
The third paragraph regarding the aetiology of liver disease (lines 142-152 pag. 4) should be moved up, inside the second paragraph, where the results regarding the demographical and clinical data are presented. Please follow the order as presented in the Table 1 and, if possible, separate “Patient and Tumour Characteristics” in two different sections.
Moreover, I believe there are some typos, in particular in line 118 pag. 3 and in line 165 pag. 5 (“Error! Reference source not found.”). Please check the manuscript.
Figure 1: this part should be put before discussing HCC, i.e. after the demographical and clinical considerations, as presented in the Table 1.
Table 1: please check the percentages of aetiologies in patients under surveillance since the total is not equal to 100%. After that, please check the results of comparison between the two groups regarding this data and correct the text. Also the percentages of the BCLC stage of patients under surveillance are incorrect and should be checked. Finally, please use the same font for the entire Table (in particular, the WHO performance status).
Figure 2: please modify the legend of the figure by specifying that the numbers 0-4 refer to the WHO performance status.
The section, 3.3. Overall survival should be merged with the previous section (3.2. HCC Treatment).
The sections 3.4 and 3.5 should also be merged in one section titled “Comparison of cirrhotic and non-cirrhotic patients in the non-surveillance cohort”. Moreover, if the Editor agrees, since the results regarding this data are not statistically significant (with exception of size and alcohol aetiology), both Table 2 and Figure 3 can be moved to Supplemental material.
DISCUSSION
In my opinion, the topics of the discussion are very beautiful but overall the discussion is too long. The authors must decide to reduce the text which otherwise becomes difficult to read. In particular, lines 284-311 pag. 10-11 can be reduced and moved after the discussion of the analysis of survival data, following the same order of the results.
Lines 339-340: the sentence “Other frailty scores are available, such as the Liver Frailty Index, to specifically evaluate functional status in this disease group.[22]” is redundant and not necessary.
In addition, the COVID-19 pandemic has had a tremendous impact on both the management and treatment of patients with HCC, causing the rescheduling of screening exams and the shift of liver cancer therapy toward nonsurgical procedures [doi: 10.3390/ijms24021091]. Since the present study was performed between between August 2012 and December 2021, have the Authors experienced a significant decrease in the number of new HCCs diagnosed during the pandemic period? Moreover, HCCs diagnosed following the COVID-19 outbreak were significantly larger and were treated differently from before? Please cite this article and, if these considerations were not performed, please state it as a limitation for the study.
Finally, limitations should be the last section of Discussion, thus Authors should move this paragraph down.
Round 2
Reviewer 2 Report
The authors have adequately addressed my remark